# Fleece-Lined Whelping Pools Associated with Reduced Incidence of Canine Hip Dysplasia in a Guide Dog Program

**DOI:** 10.3390/ani15020152

**Published:** 2025-01-09

**Authors:** Lynna C. Feng, Alexandra Philippine, Elizabeth Ball-Conley, Sarah-Elizabeth Byosiere

**Affiliations:** 1Guide Dogs for the Blind, San Rafael, CA 94903, USA; 2Animal Dairy and Veterinary Sciences, Utah State University, Logan, UT 84322, USA

**Keywords:** dog, orthopedic, hip dysplasia, development, environment

## Abstract

Canine hip dysplasia (CHD) is a complex orthopedic disease. Genetic selection can be effective in reducing the incidence rate of CHD, but environmental factors may also impact the expression of this developmental condition. The objective of this retrospective research was to investigate the relationship between the type of substrate in the whelping pool (used from birth to 3 weeks of age) and CHD. Records from 5649 dogs (Labrador Retrievers, Golden Retrievers, and Labrador–Golden Retriever crosses) at Guide Dogs for the Blind, USA, were analyzed. From 1 July 2012 to 28 November 2015, 2785 dogs were whelped in pools lined with newspaper. From 5 March 2016 to 29 December 2019, 2864 dogs were whelped in pools lined with fleece. The results suggested that dogs whelped in pools lined with fleece had a lower risk for CHD diagnosis but similar PennHIP scores compared to those whelped in pools lined with newspaper. Golden Retrievers had the highest risk of CHD diagnosis and highest PennHIP scores, followed by Labrador–Golden Retriever crosses and finally Labrador Retrievers. These correlational findings suggest that environmental factors, such as the substrate type, during the neonatal period may play a role in the development of future orthopedic disease.

## 1. Introduction

Canine hip dysplasia (CHD) is characterized by the abnormal development and laxity of the hip joint. This malformation can significantly accelerate the expected wear on the joint over time, increasing the risk of secondary orthopedic diseases and negatively impacting a dog’s welfare. Dogs with CHD experience progressive osteoarthritis, causing chronic pain and lameness as the disease progresses. The socioeconomic impacts of CHD management and treatment are also significant. Surgical repairs are estimated to cost between USD 1500 and USD 7000 depending on the procedure performed, and this can be even higher depending on the geographical location. While CHD can have a significant impact on pet dogs and their owners, the broader effects may be even greater in dogs bred and trained for working roles, ranging from police and military work to mobility aids for individuals with disabilities. Often, dogs bred for these roles are rejected from their respective programs due to poor-quality hips, often after many months of training and rearing expenses have been invested [1].

First described by Dr. Gerry Schnelle in 1935 [2], CHD was originally presumed to be a rare condition. Today, CHD is a commonly diagnosed orthopedic disease in dogs, affecting anywhere from 0 to 73.4% of dogs, depending on the breed [3,4]. Often more prevalent in medium- and large-breed dogs, CHD expression is polygenic and multifactorial, with both genetic and environmental factors. In assessing 501 client-owned dogs from 103 litters, Krontveit et al. [5] evaluated the housing- and exercise-related risk factors associated with the development of CHD as determined by radiographic evaluation in Newfoundlands, Labrador Retrievers, Leonbergers, and Irish Wolfhounds in Norway. Overall, 24.6% of dogs were diagnosed with CHD, with 36% of Newfoundlands, 20.3% of Labrador Retrievers, 25% of Leonbergers, and 9.5% of Irish Wolfhounds being affected. An increased risk of CHD was observed for dogs whose breeder and/or owner reported, via a questionnaire, the daily use of stairs from birth until three months of age. A decreased risk of CHD was observed for dogs whose breeder and/or owner reported daily off-leash exercise in the park from birth until three months of age. Thus, a critical time for puppies and future CHD expression related to environmental impacts may be from birth until 3 months of age.

Interestingly, Krontveit et al. [5] did not find that any housing-related variables, specifically related to indoor and outdoor flooring surfaces, were associated with increased CHD expression. This result is contrary to those previously reported by van Hagen et al. [6] in a cohort of 1733 Boxers from 325 litters in the Netherlands. Cumulative hazard rates for clinical signs of CHD (cCHD) were estimated from 7 weeks to 8 years of age and reported to be 8.5%. Puppies that were kept on slippery floors (e.g., newspaper or tarpaulin) as neonates when housed with the dam were 1.6 times more likely to develop cCHD compared to neonates kept on non-slippery floors (e.g., carpet, rubber, blankets, sawdust, straw). Therefore, it is possible that, for certain breeds, housing-related environmental factors are of importance—specifically the flooring type—during preweaning development.

Given the critical period of 0–3 months of age in which Krontveit et al. [5] observed an exercise-related risk of developing CHD, and the housing-related risk of the flooring type observed by van Hagen and colleagues [6], the current study aimed to further investigate the relationship between CHD and the flooring substrate. Specifically, Guide Dogs for the Blind, USA (GDB), set out to retrospectively assess whether a protocol change from slippery whelping pen flooring (newspaper) to non-slippery fleece flooring (STERI-LON^®^ By The Roll Fleece with a non-slip backing) would be associated with a reduced rate of CHD diagnoses.

## 2. Methods

The current study represents a retrospective and correlational investigation in which historical records were reviewed and analyzed. All aspects of this investigation were approved and monitored by an internal research committee at Guide Dogs for the Blind (GDB).

### 2.1. Subjects

The historical records assessed in the present investigation concerned dogs that were bred at GDB, a non-profit guide dog organization headquartered in San Rafael, CA, USA. All puppies were whelped and reared at the organization’s breeding and training campus in San Rafael, CA, USA, between 1 July 2012 and 29 December 2019. The final analysis included historical records from dogs who were placed in GDB’s puppy raising program at around eight weeks of age. However, in both the newspaper and fleece groups, various subjects were excluded. In the newspaper group, these included 222 stillborn/neonatal deaths, 56 dogs that were donated and/or exchanged with other guide dog or assistance dog organizations, and 18 dogs who were released from GDB’s program prior to being placed in puppy raising. In the fleece group, this included 233 stillborn/neonatal deaths, 80 dogs that were donated and/or exchanged with other guide dog or assistance dog organizations, and 21 dogs who were released from GDB’s program prior to being placed in puppy raising.

### 2.2. Care and Diagnosis of Orthopedic Disease

Due to the important possible benefits of utilizing a non-slip surface in the whelping pool (fewer diagnoses of CHD), GDB elected against a traditional randomized controlled treatment design. Instead, a historical control group was used for comparison. For a detailed discussion of the possible limitations associated with this type of methodological design, please see the Discussion. Prior to December 2015, the puppies were whelped and raised with their littermates in plastic clamshell pools lined with flat newspaper (newspaper group). Beginning in March 2016, all litters were whelped and raised in plastic clamshell pools lined with anti-microbial veterinary fleece called STERI-LON^®^ (fleece group). In both groups, the litters were moved out of the pools onto the epoxy flooring of the whelping kennel stall at around three weeks of age and were housed on this surface until leaving the breeding campus.

At approximately six weeks of age, the puppies receive a physical exam by a veterinarian, and large litters are separated into smaller groups of two to four puppies per kennel run. Around nine weeks of age, the puppies are driven or flown offsite to be raised in volunteer homes across ten states in the western United States. The volunteers utilize their local veterinary services for all preventative care and basic treatment. Any puppy who presents with potential signs of pain related to orthopedic disease (such as lameness or unusual gait) receives further diagnostics as recommended by the local veterinary professional, which may include the radiographic assessment of the hip joints. Around sixteen months of age, young dogs who meet the behavioral and temperamental criteria travel to one of two program sites (Boring, Oregon and San Rafael, CA, USA) for the next phase of their training and assessment for breeding. These dogs receive a radiographic evaluation by staff veterinarians at GDB to identify subclinical evidence of hip dysplasia. The subset of dogs being considered for breeding receive additional radiographs that are submitted to Antech Imaging Services for PennHIP testing.

### 2.3. Statistical Analysis

Descriptive statistics are reported as counts and medians ± interquartile ranges for the non-parametric continuous PennHIP distraction index scores. As the dogs in this study belonged to a highly related population, a Bayesian statistical approach was employed for inferential modeling to account for the impact of relatedness on the canine hip dysplasia (CHD) diagnosis and PennHIP scores. Bayesian regression models were conducted using the public-domain language R (v4.3.3) [7] and the brm() function in the Bayesian package brms [8]. The models included a presumed quantitative genetic contribution.

A logistic regression was used to model the risk of disease (diagnosis of CHD present or absent), where the probability of disease was defined as $p_{ijkl}$ for the *l*-th dog in the *i*-th breed group (Labrador Retriever, Golden Retriever, Labrador–Golden Retriever cross) of the *j*-th sex (female, male) and having been in the *k*-th treatment group (newspaper, fleece). The logit of this probability was defined as $\theta_{ijkl}=log\left[p_{ijkl}/\left(1-p_{ijkl}\right)\right]$, modeled as a function of the breed, sex, group, and quantitative genotype, where $a_{ijkl}$ is the additive genetic contribution to the risk of disease for the *l*-th (*l* = 1, 2, 3, …, 5649) dog and $e_{ijkl}$ is an unknown random residual particular to the *l*-th dog:$$\theta_{ijkl}={breed}_{i}\;+\;{sex}_{j}+{group}_{k}+a_{ijkl}+e_{ijkl}$$

A linear regression was used to model a dog’s PennHIP distraction index score. As larger scores are suggested to be associated with an increased risk of degenerative joint disease [9] and osteoarthritis [10], the dog’s maximum PennHIP score across the right and left hip was used for modeling to best represent the risk of disease in any hip joint for that individual. The PennHIP scores were once again modeled as a function of the breed, sex, group, and quantitative genotype:$${PH}_{ijkl}={breed}_{i}\;+\;{sex}_{j}+{group}_{k}+a_{ijkl}+e_{ijkl}$$

The heritability of the disease risk and PennHIP scores can be estimated as $h^{2}={\;\sigma }_{a}^{2}/({\;\sigma }_{a}^{2}+1)$. For both models, the simulation was conducted across 4 chains, drawing 25,000 total samples, with a “burn-in” process of 5000 samples followed by thinning to every 20th sample. The convergence of the process was visualized through trace plots of all the unknown values and the computation of the Gelman-Rubin statistic for convergence (Rhat) below 1.05 [11].

## 3. Results

The complete data set (Appendix A) consisted of 5649 dogs (51.53% male, 48.47% female), including dogs with (*n* = 64) and without (*n* = 5585) a diagnosis of canine hip dysplasia (CHD). Of these, there were 4446 Labrador Retrievers (LAB), 561 Golden Retrievers (GLD), and 642 mixed breeds (multi-generational Labrador–Golden Retriever crosses; LGX) from a total of 935 litters. In total, 2785 dogs belonged to the newspaper group (whelped between 1 July 2012 and 28 November 2015), with 54 diagnosed cases of CHD (1.94%). The remaining 2864 dogs were in the fleece group (whelped between 5 March 2016 and 29 December 2019), with 10 diagnosed cases of CHD (0.35%). The radiographic evaluation of the hip joints was conducted for 70.76% of the dogs (1994/2785 newspaper group; 2003/2864 fleece group). The two treatment groups were approximately equal in terms of the breed group distribution and sex ratio (Table 1).

The visual inspection of the incidence rates of hip dysplasia by breed and whelp year suggested a difference between the newspaper and fleece groups (Figure 1).

### 3.1. CHD Diagnosis Modeling

The Bayesian logistic regression modeling of CHD diagnosis supported the visually identified differences according to the breed group and treatment group (Table 2). The model simulation converged successfully, with all Rhat values equal to 1.00; however, the effective sample sizes were approximately half the true number of samples due to some correlations between the observations (related individuals in the sample) for all effects except sex, where the approximated effective sample size was approximately equal to 4000, i.e., the actual sample size. Despite this, the dogs in the fleece group were found to be less likely to be diagnosed with CHD compared to the newspaper group.

Golden Retrievers were most likely to be diagnosed with CHD; Labrador–Golden Retriever crosses were also more likely than Labrador Retrievers to be diagnosed with CHD (Figure 2). No sex differences were identified by the model.

### 3.2. PennHIP Distraction Index Modeling

The Bayesian linear regression modeling successfully simulated the PennHIP distraction index scores (Table 3). All Rhat convergence scores were 1.00, meeting the convergence criteria. The effective sample sizes were close to the true simulation sample size of 4000.

Golden Retrievers had higher PennHIP scores than Labrador Retrievers (on average, 0.06 units higher). In approximately 95% of the model simulations, Labrador–Golden Retriever cross dogs also had higher PennHIP scores than Labrador Retrievers (on average, 0.04 units higher). While no sex difference was identified in the risk of CHD diagnosis, males were found to have lower PennHIP scores than females by an average of 0.04 units (Figure 3).

### 3.3. Heritability

As a byproduct of Bayesian regression modeling using the brms package, the heritability of the disease risk and PennHIP scores was estimated for both models. The heritability of CHD diagnosis was 0.237 (90% credible interval 0.075–0.419). The heritability of the PennHIP distraction index scores was 0.001 (90% credible interval 0.000–0.003).

## 4. Discussion

The results of the present investigation highlight the significant relationship between canine hip dysplasia (CHD) diagnosis and the type of substrate used in the whelping pool, particularly in puppies from whelping to approximately three weeks of age. According to the CHD model, puppies whelped in fleece-lined pools were found to have a decreased risk of CHD diagnosis compared to puppies whelped in newspaper-lined pools. While the fleece substrate type was associated with a decreased risk of CHD diagnosis, according to the CHD model, the risk of CHD diagnosis did vary by breed. In particular, the risk was higher for Golden Retrievers and Labrador–Golden Retriever crosses than Labrador Retrievers, with no difference observed between males and females. Of note, while the logistic regression modeling of the risk of CHD was successful, correlations between individuals resulted in an approximated effective sample size of only half the true sample, with very wide credible intervals for the model coefficients. A larger sample would be required to determine meaningful relative risk estimates.

It is important to emphasize that the rates of CHD in this population of guide dogs was already low, ranging from 0.35%, or 10 out of 2864 dogs (for those in the fleece group), to 1.94%, or 54 out of 2785 dogs (in the newspaper group). While the incidence rates of CHD vary and are difficult to extrapolate and compare across studies, compared to the Orthopedic Foundation of Animals (OFA), the percentage of dysplastic Labrador Retrievers and Golden Retrievers in the general population is substantially greater, at 11.5% (out of 313,105 evaluations) and 19.5% (out of 184,214 evaluations), respectively. Additionally, the heritability of CHD diagnosis was estimated to be 0.237 in our sample, suggesting low to moderate heritability. This is consistent with the ranges of heritability for hip dysplasia (between 0.1 and 0.6) published in the literature [12,13,14,15].

Interestingly, while the substrate type was associated with the risk of CHD in the CHD model, a key difference was observed when compared to the PennHIP model. No differences were detected in the PennHIP model for the scores of the dogs in the fleece group compared to the newspaper group. It is possible that this non-significant finding is related to a sampling bias within this population, as only adult dogs being considered for the breeding colony were scored using PennHIP assessments. The PennHIP model only included 429 total dogs, with a heavy skew towards females (342; 79.7%) relative to the full sample. Although the breed groups were similarly represented in the PennHIP sample relative to the full sample (Labrador Retrievers making up 80.7% of the PennHIP subset vs. 78.7% of the full sample; Golden Retrievers making up 10.5% of the PennHIP subset vs. 9.9% of the full sample; and Labrador–Golden Retriever crosses making up 8.8% of the PennHIP subset vs. 11.4% of the full sample), with a total PennHIP sample of 429, this only included 45 Golden Retrievers (13 of these being male) and 38 Labrador–Golden Retriever crosses (three of these being male). Overall, there was limited variability in the PennHIP scores (90% of scores falling between 0.2 and 0.5). It is likely that the small sample and the very narrow range of PennHIP scores limited the observable impact of the model effects on the variance compared to the CHD model. This explanation is further supported by the fact that the heritability estimates for the PennHIP model were essentially 0, which is also likely a reflection of the sample rather than a true null result.

However, some consistencies between the PennHIP and the CHD model were noted. The PennHIP model predicted scores that were 0.05 units higher for Golden Retrievers than Labrador Retrievers and 0.04 units higher for Labrador–Golden Retriever crosses than Labrador Retrievers, although the coefficient for Labrador–Golden Retriever crosses had a lower 95% credible interval value of 0.00, meaning that 5% of the model iterations did not predict an increased PennHIP score for Labrador–Golden Retriever crosses compared to Labrador Retrievers. In the PennHIP model, males were observed to have lower PennHIP scores than females, by about 0.04 units. This closely matches the sex effect reported in another guide dog program, with a dog population of German Shepherd dogs, Labrador Retrievers, and Golden Retrievers, where males had an average PennHIP distraction index score that was 0.034 units lower than that of females [16].

The results of the present investigation are informative and contribute to a growing body of literature. To our knowledge, few peer-reviewed empirical studies have directly focused on assessing the relationship between the substrate type in the environment during early preweaning development and CHD diagnoses. One study, perhaps the most comparable, conducted by van Hagen et al. [6] in a cohort of 1733 Boxers, found that puppies housed on slippery floors, including newspaper, between 0 and 3 months of age were more likely to develop cCHD compared to those housed on non-slippery floors. In line with van Hagen et al. [6], our results provide additional evidence that the substrate type—specifically non-slip flooring utilized during preweaning in puppies—is correlated with reduced CHD diagnoses. Of note, the puppies in our study were only housed on newspaper or fleece while in the whelping pool (until around 3 weeks of age); after this age, all puppies were housed on epoxy flooring with limited traction compared to the antimicrobial fleece. Future research is needed to explore whether there could be additional protective effects of the fleece substrate beyond 3 weeks of age.

As is true for many applied investigations, the present study was subject to limitations that should be considered when interpreting and extrapolating the results. While the demographic distributions across both samples were relatively equal, it is still possible that this methodological design was subject to various biases beyond the researchers’ control. The present study was retrospective in nature, comparing records across two different time points. The results are correlational, not causal, and require caution to avoid overinterpretation. This methodology was utilized given (1) the possibility that a change in substrate type—specifically from newspaper to a non-slip fleece—may have resulted in possible health benefits (reduction in orthopedic disease) and (2) that GDB “continually evaluates and advances internal programs and protocols using the latest scientific, technological, and real-world insights to champion standards of excellence in the care, welfare, and overall well-being of our dogs” [17].

Due to the methodology used, it is possible that the change in CHD diagnoses between the two conditions was due to ongoing selective pressures and breeding practices, rather than the substrate type during pre-weaning. However, when evaluating the changes in CHD diagnoses across years, regardless of the substrate condition, there did not appear to be a steady decline in CHD diagnoses. Rather, a visible drop-off was apparent, especially in Labrador Retrievers (Figure 1a), with Golden Retrievers (Figure 1b) and Labrador–Golden Retriever crosses (Figure 1c) following a similar pattern, with some fluctuation, as expected with the smaller sample sizes.

While selection and breeding pressures may not solely explain the differences between the two substrate conditions, other environmental impacts may have contributed to the change in CHD diagnoses observed. These include ontogenetic influences, which are expected to differ across an individual’s lifespan, including, but not limited to, diet, learning history, and geographical location. For example, this study took place at an active and operating organization with many staff members. It is likely that the staff varied over the years, including staff directly involved with canine neonatal care and canine veterinary care. This limitation is impossible to quantify but should be carefully considered, among others, when generalizing and/or applying the results to other settings.

Finally, it is important to emphasize that, for some dogs, radiological screening may not have been available. While all dogs that arrive for training or breeding evaluations are screened radiologically, puppies who are released from the program before this time may or may not have had radiological screening. This means that it is possible that the sample analyzed was not completely representative of the population at Guide Dogs for the Blind.

## 5. Conclusions

In conclusion, a significant correlational relationship was observed between canine hip dysplasia (CHD) diagnosis and the type of substrate used in the whelping pool (from birth to 3 weeks). Puppies whelped at Guide Dogs for the Blind, USA, whelped in fleece-lined pools, were found to have a decreased risk of CHD diagnosis compared to puppies whelped in newspaper-lined pools. While no sex differences were noted, there was a difference across the breeds evaluated. In particular, Golden Retrievers and Labrador–Golden Retriever crosses had a higher risk of CHD diagnosis than Labrador Retrievers.

To the authors’ knowledge, this study represents one of the few, and one of the largest, investigations assessing the use of a non-slip substrate, namely fleece, in whelping pools during preweaning and its association with CHD diagnoses. The results reported are informative for a variety of professionals, particularly those directly involved in canine breeding practices. Given the single population assessed, it is recommended that future research more broadly investigate the environmental impacts of substrate surfaces on orthopedic health in puppies, across multiple breeds (including both large- and small-breed dogs) and populations. Finally, additional replication will be critical in further establishing our understanding of this polygenic and multifactorial disease.

## Figures and Tables

**Figure 1 animals-15-00152-f001:**
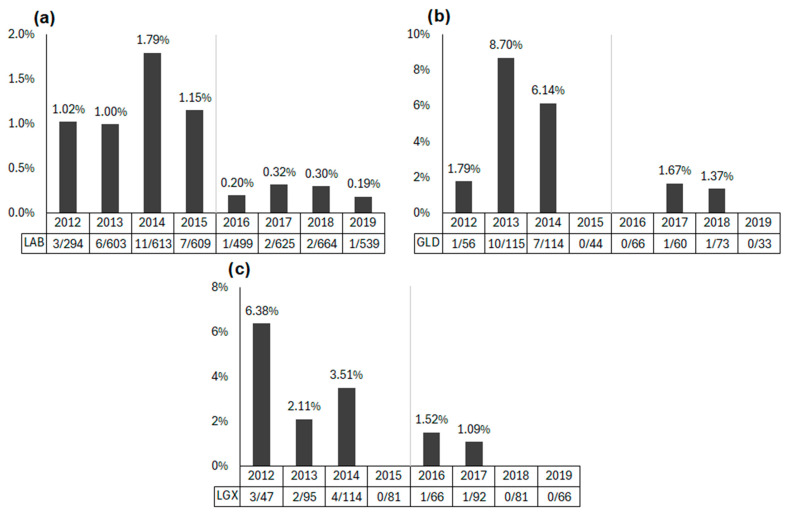
Three graphs of hip dysplasia incidence rate by dogs’ whelp year split by breed group ((**a**) Labrador Retriever, (**b**) Golden Retriever, (**c**) Labrador–Golden Retriever cross). A vertical line between 2015 and 2016 on each graph reflects the two treatment groups (newspaper for puppies whelped between 2012 and 2015 and fleece for puppies whelped between 2016 and 2019). Data labels below each year show the proportion of dogs diagnosed with hip dysplasia out of all dogs of that breed who were whelped that year.

**Figure 2 animals-15-00152-f002:**
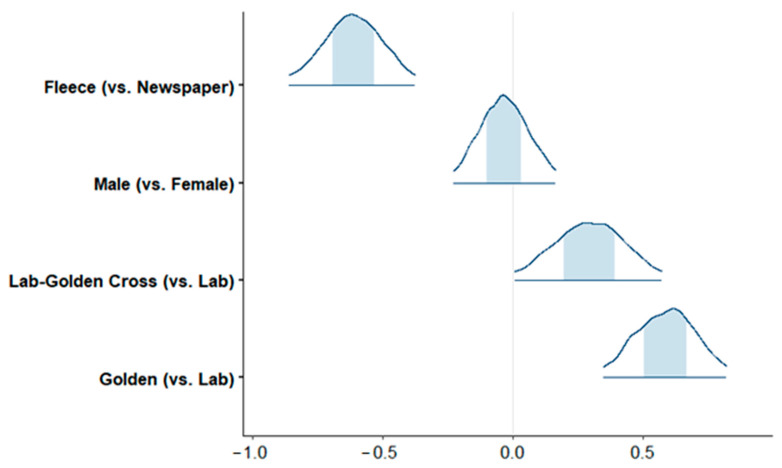
CHD logistic regression coefficients with 95% posterior distributions of regression coefficients, with the shaded area reflecting the middle 50% of the simulations.

**Figure 3 animals-15-00152-f003:**
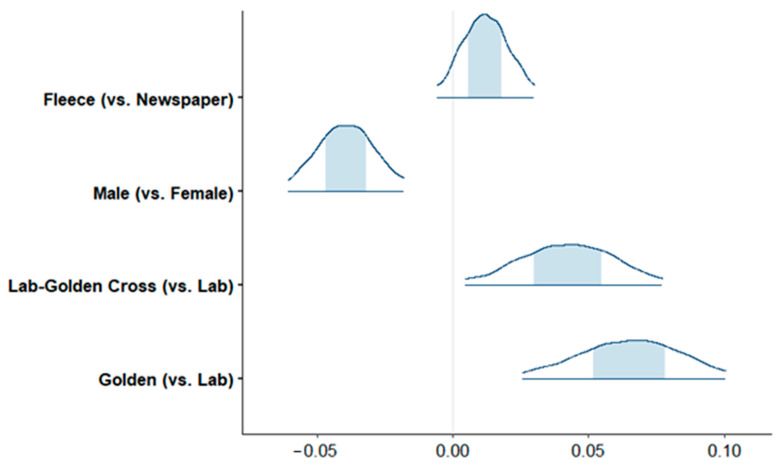
PennHip linear regression coefficients with 95% posterior distributions of regression coefficients, with the shaded area reflecting the middle 50% of the simulations.

**Table 1 animals-15-00152-t001:** Number of subjects and incidence of hip dysplasia by breed group and treatment group (LAB: Labrador Retriever, GLD: Golden Retriever, LGX: Labrador–Golden Retriever cross).

		LAB	GLD	LGX
Sex (male; female)	Newspaper	1113 M; 1006 F	157 M; 172 F	162 M; 175 F
	Fleece	1223 M; 1104 F	124 M; 108 F	132 M; 173 F
Radiographic evaluation of hip joints (*n*, %)	Newspaper	831 M; 775 F (75.79%)	85 M; 97 F (55.32%)	92 M; 114 F (58.16%)
	Fleece	857 M; 836 F (72.75%)	73 M; 58 F (56.47%)	77 M; 102 F (58.69%)
Hip dysplasia (*n*, %)	Newspaper	27 (1.27%)	18 (5.47%)	9 (2.67%)
	Fleece	6 (0.26%)	2 (0.86%)	2 (0.66%)
PennHIP scores (*n*, median and IQR)	Newspaper	0.33 (0.27–0.40) (38 M; 150 F)	0.38 (0.33–0.49) (5 M; 19 F)	0.34 (0.27–0.37) (1 M; 11 F)
	Fleece	0.34 (0.28–0.41) (32 M; 126 F)	0.36 (0.26–0.51) (8 M; 13 F)	0.42 (0.35–0.50) (3 M; 23 F)

**Table 2 animals-15-00152-t002:** Logistic regression coefficients modeling dogs’ risk of CHD diagnosis with 95% credible intervals, estimated error, Rhat measure of convergence, and estimated effective sample size.

	Regression Coefficient (95% CI)	Est. Error	Rhat	Effective Sample Size
Group-Level Effects				
sd (Intercept)	**5.95 (1.81, 18.53)**	4.65	1.00	2035
Population-Level Effects				
Intercept	**−13.22 (−37.44, −5.62)**	8.99	1.00	2050
Golden Retriever (vs. Labrador Retriever)	**4.26 (1.17, 12.53)**	2.91	1.00	2588
Labrador–Golden Retriever Cross (vs. Labrador Retriever)	**2.73 (0.15, 8.7)**	2.17	1.00	2695
Fleece (vs. Newspaper)	**−2.74 (−8.44, −0.81)**	1.96	1.00	2394
Male (vs. Female)	−0.36 (−2.34, 0.78)	0.75	1.00	4012

Regression coefficients in bold where the 95% credible intervals do not include 0.

**Table 3 animals-15-00152-t003:** Linear regression coefficients modeling dogs’ PennHIP distraction index scores with 95% credible intervals, estimated error, Rhat measure of convergence, and estimated effective sample size.

	Regression Coefficient (95% CI)	Est. Error	Rhat	Effective Sample Size
Group-Level Effects				
sd (Intercept)	**0.04 (0.01, 0.06)**	0.01	1.00	3683
Population-Level Effects				
Intercept	**0.36 (0.34, 0.37)**	0.01	1.00	3891
Golden Retriever (vs. Labrador Retriever)	**0.06 (0.03, 0.10)**	0.02	1.00	3998
Labrador–Golden Retriever Cross (vs. Labrador Retriever)	0.04 (0.00, 0.08)	0.02	1.00	3662
Male (vs. Female)	**−0.04 (−0.06, −0.02)**	0.01	1.00	3984
Fleece (vs. Newspaper)	0.01 (−0.01, 0.03)	0.01	1.00	3604
Family specific parameters				
sigma	0.08 (0.07, 0.08)	0.00	1.00	3677

Regression coefficients in bold where the 95% credible intervals do not include 0.

## Data Availability

The original contributions presented in this study are included in the article. Further inquiries can be directed to the corresponding author.

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
