# Peer review of "Fleece-Lined Whelping Pools Associated with Reduced Incidence of Canine Hip Dysplasia in a Guide Dog Program"

_animals, 2025, doi:10.3390/ani15020152_

Round 1

Reviewer 1 Report

Comments and Suggestions for Authors

SUMMARY

This paper examines the relationship between the type of substratum (newspaper vs. fleece) used in whelping pools and the development of canine hip dysplasia in working dogs, specifically guide dogs for visually impaired individuals. The study identifies a correlation between the substratum utilized and the risk of canine hip dysplasia, finding a decreased risk in puppies raised in fleece whelping pools.

This research stands out for its significant social relevance, its robust sample size, appropriate methodology, thorough statistical analysis, clear results, and a well-constructed discussion that addresses limitations. However, the paper presents several critical weaknesses that require attention:

Major Corrections:

1.        Formatting:

1.1.  Please adapt the format of the bibliographic citations to meet the required standards.

1.2.  Simple summary is need it according to the format

1.3.  The paper lacks a conclusion section; please include one to summarize the findings and their implications (this is the reason for considering the paper into major revisions, is it necessary to correct the conclusions according to the format of this journal).

Minor Corrections:

2. In lines 73-76, you mention that the study was approved by an internal research committee at GBD. If any documentation is available, it is essential to mention it in the "Institutional Review Board Statement" section.

3. Although the study’s limitations are relatively well explained, please clarify in the "Subjects" section whether inclusion and exclusion criteria were applied to the data analyzed.

Additional Feedback:

  1. Is the topic original or relevant in the field, and if so, why?

Yes, the topic is both original and relevant. As stated in the paper, there is limited literature addressing this issue, which is critical for improving the health of companion animals and fostering societal well-being.

  1. What does it add to the subject area compared to other published material?

This study establishes a correlation between the type of substratum used in whelping pools and the development of hip dysplasia in working dogs trained as guides for visually impaired individuals.

  1. What specific improvements could the authors consider?

The authors should ensure proper formatting of the paper, particularly the bibliographic references.

  1. Are the conclusions consistent with the evidence and arguments presented, and do they address the main question posed?

The paper lacks a conclusions section, which must be included to address the main research question and summarize the evidence.

  1. Are the references appropriate?

It could be improved, the references are adequate; however, the addition of more sources could further strengthen the paper.

Author Response

Comment 1:  Please adapt the format of the bibliographic citations to meet the required standards.

Response 1: The citations (in text and reference list) have now been formatted using the multidisciplinary digital publishing institute style file for Zotero.  

Comment 2:  Simple summary is need it according to the format

Response 2: We have now included a simple summary for the manuscript.

Comment 3:  The paper lacks a conclusion section; please include one to summarize the findings and their implications (this is the reason for considering the paper into major revisions, is it necessary to correct the conclusions according to the format of this journal).

Response 3: Thank you for identifying that a conclusion section was missing. We have now added additional text that broadly overviews the results and reformatted the final paragraph of the manuscript to create this new section.

Comment 4: In lines 73-76, you mention that the study was approved by an internal research committee at GBD. If any documentation is available, it is essential to mention it in the "Institutional Review Board Statement" section.

Response 4: As this project evaluated historical data, specifically veterinary records, no additional documentation was required.

Comment 5: Although the study’s limitations are relatively well explained, please clarify in the "Subjects" section whether inclusion and exclusion criteria were applied to the data analyzed.

Response 5: Thank you for asking this critically important question. The final sample consisted of most dogs; however, some exclusions were implemented. These are now identified in the manuscript for both treatment groups.

Comment 6: The addition of more sources could further strengthen the paper.

Response 6: We appreciate the reviewer’s concern regarding the limited number of citations presented. Unfortunately, the available peer-reviewed literature on this topic is extremely limited. We believe we have included most, if not all, relevant citations to the manuscript and its scope but are open to reviewing and including additional peer-reviewed literature if any key citations have been missed.

Reviewer 2 Report

Comments and Suggestions for Authors

Canine hip dysplasia is an ongoing problem in dogs. In the introduction, the authors sufficiently describe the history of the disease and its incidence with respect to dog breeds.
The material and methods are well described. The selection of puppies for Guide Dogs for the Blind is described in detail. The number of dogs included in the study is very large. A limitation certain is that only 2 breeds of dogs and mixedbreed were used for the study.Two groups were separated and Puppies were maintained until the age of 3 weeks on two different substrates.
The results of the study are very interesting and indicate a significant effect of the substrate on the incidence of dysplasia. As the authors rightly note, this is a direction of research that has not been exposed before. The discussion is appropriate.
The literature is appropriate.
The authors could be tempted to make some hypotheses about the effect of substrate on the developing and plastic hip joint as well as the plastic pelvis during this period. Perhaps the reduced amount of microtrauma associated with the lack of slippage is relevant?
The work indicates how important environmental conditions are for the potential development of dysplasia.

With Table 1, explain the abbreviations M- male F- female in the table description

In table 2 Lab- labrador retriver

Author Response

Comment 1: With Table 1, explain the abbreviations M- male F- female in the table description. In table 2 Lab- labrador retriever

Response 1: Thank you for identifying this oversight. We have now explained the abbreviations in the table descriptions or in the table itself.

Reviewer 3 Report

Comments and Suggestions for Authors

The reviewed manuscript, entitled: Fleece-lined whelping pools associated with reduced incidence 1 of canine hip dysplasia in a guide dog program, seems to be a valuable contribution for Canine hip dysplasia multifactorial aetiology understanding. As well known, the environmental factors play crucial role in animal postnatal development since birth to the 3 months old dogs. The authors studied these factors and brought to the light very important conclusions. The abstract and summary is well organized and fully described the study. The introduction describes the current status of knowledge in this field of science and clearly defines the aim of the study. The material and methods used in study are adequate. The number of investigated individuals is more than enough, the breeders care and veterinary care organization was optimal for obtaining the reasonable results. The results are clearly presented and statistically analysed. These gave the essential interpretation and discussion. The conclusion is clear.

I have no remarks to the scientific work quality presented by authors. I suggest to accept the manuscript in current form.

Author Response

Response 1: We thank the reviewer for their positive views on the submitted manuscript. 

Round 2

Reviewer 1 Report

Comments and Suggestions for Authors

Thank you for addressing the corrections provided by this reviewer in an appropriate and satisfactory manner.

Author Response

We thank the Reviewer for their willingness to re-review the manuscript and their final comments regarding the changes made.